# Distribution and Phylogenetic Diversity of *Synechococcus*-like Cyanobacteria in the Late Autumn Picophytoplankton of the Kara Sea: The Role of Atlantic and Riverine Water Masses

**DOI:** 10.3390/plants14172614

**Published:** 2025-08-22

**Authors:** Tatiana A. Belevich, Irina A. Milyutina, Andrey B. Demidov, Olga V. Vorob’eva, Alexander A. Polukhin, Sergey A. Shchuka, Aleksey V. Troitsky

**Affiliations:** 1Biological Faculty, Lomonosov Moscow State University, 119234 Moscow, Russia; olvorobieva@rambler.ru; 2Belozersky Institute of Physico-Chemical Biology, Lomonosov Moscow State University, 119992 Moscow, Russia; iramilyutina@yandex.ru; 3Shirshov Institute of Oceanology Russian Academy of Science, 117997 Moscow, Russia; demspa@rambler.ru (A.B.D.); aleanapol@gmail.com (A.A.P.); s_shchuka@mail.ru (S.A.S.)

**Keywords:** *Synechococcus*, marine cyanobacteria, *petB* gene, metagenomics, Kara Sea

## Abstract

Increased Atlantic water transport and river discharge are more pronounced effects of global warming at high latitudes. Both phenomena may lead to changes in the species composition of small-celled algae populations in marine ecosystems, as well as to the emergence of new species. This study investigated the spatial distribution of picocyanobacterial (PC) abundance and the phylogenetic diversity of PC *Synechococcus* in the Kara Sea. PC abundance varied from 2 to 88 cells mL^−1^ and increased with warming temperatures and decreasing salinity caused by river water influence. The contribution of *Synechococcus* to the total picophytoplankton biomass was low (<16%). The *Synechococcus* community was characterized at deep taxonomic level using amplicon sequencing targeting the *petB* gene. Diversity was low, revealing only *Synechococcus* subcluster 5.1 polar lineages I and IV, and euryhaline subcluster 5.2. *Synechococcus* subcluster 5.1.I represented on average 97% of the total reads assigned to cyanobacteria. For the first time, the presence of estuarine *Synechococcus* subcluster 5.2 was documented as far north as 82° N. Modified Atlantic water was the main source of cyanobacteria in the Kara Sea, followed by river discharge. Our study contributes to the understanding of PC sources in the Kara Sea and allows for the further monitoring of PC distribution and evolution.

## 1. Introduction

Unicellular picocyanobacteria [<2 µm] belong to the functional class of picophytoplankton, the smallest algae that contribute substantially to both the total phytoplankton biomass and primary production in marine ecosystems, especially in the oligotrophic waters of the Arctic Ocean, where they account for up to 55–90% of the total photosynthetic biomass and carbon production [1,2,3]. Global analyses show that in lower and temperate latitudes, picocyanobacteria (PC), presented by the genera *Prochlorococcus* and *Synechococcus*, contribute significantly to marine picophytoplankton [4,5]. In contrast, picoeukaryotes dominate polar marine ecosystems, with *Prochlorococcus* considered to be virtually absent and *Synechococcus* more ubiquitous and found in low abundances in the Arctic Ocean [6,7,8,9].

The polar regions have become a focus for research in recent decades due to the alarming effects of global warming [10,11,12]. In the Arctic marine environment, the surface waters are warming and becoming fresher as a result of the melting of multi-year sea ice and increased river discharge [13,14]. These changes may dramatically alter the structure of primary producer communities. Recently, *Prochlorococcus marinus* was identified in the polar waters of the Spitsbergen shelf, the Fram Strait [15]. The authors linked this fact to the process of the Atlantification of the Arctic, which involves the increased transport of Atlantic water masses through the Fram Strait towards the Arctic Ocean.

Among the typical Arctic marginal seas, the Kara Sea stands out for its strong freshwater outflow, the volume of which exceeds 1200 km^3^ per year and is mainly determined by the large Ob and Yenisei rivers [16,17]. River discharge enters the shelf sea from a river mouth, forming a so-called “river plume” that spreads and mixes with ambient saline sea water, forming a freshened water mass in the upper Kara Sea [18,19]. Freshwater is found throughout the surface layer of the Kara Sea, except in a few areas near the northern part of the Novaya Zemlya Archipelago. The Kara Sea is also influenced by warm Atlantic waters coming from the west through the Kara Gate Strait and from the northwest in the area of the St Anna Trough. Thus, the waters of the Kara Sea result from the mixing of ambient seawater (of Atlantic origin) with river runoff [20,21].

In polar waters, the origin of PC may be either autochthonous or allochthonous [5,8,22]. River flow and North Atlantic waters can serve the main sources of allochthonous cyanobacteria in the Kara Sea. Under current global warming trends, the volume of freshwater outflow and warm Atlantic waters will continue to increase, leading to changes in phytoplankton composition and size structure, and an increase in the abundance of PC in Arctic phytoplankton is expected [23]. Therefore, studying the recent spatial distribution and composition of PC in polar waters may enable us to evaluate the contribution of river runoff and altered Atlantic water to allochthonous cyanobacterial populations. This would provide baseline data for the Kara Sea, from which future changes could be monitored. Arctic seas are difficult to access, and observations are scarce in these areas, so the data obtained could help to develop models for high-latitude systems.

Picocyanobacteria *Synechococcus sensu lato* is a high phylogenetically and phenotypically diverse group [24]. Various markers have been used for the phylogenetic classification of these cyanobacteria, first of all 16S rDNA, but also the ITS rDNA region, *rpoC1*, *narB*, *ntcA*, and *petB* genes [25,26,27,28,29,30]. Dufresne et al. (2008) analyzed the complete genomes of 11 marine *Synechococcus* isolates and classified them into three subclusters, 5.1, 5.2, and 5.3 [25]. Other authors analyzing different markers have proposed various nomenclatures for marine *Synechococcus*, recognizing up to 27 clades within subclusters 5.1–5.3 [31]. Ahlgren and Rocap, based on multiple gene loci, recognized more than 30 marine-cultured *Synechococcus* clades in three subclusters [32]. Coutinho et al. (2016) [33], based on phylogenomic reconstruction, segregated *from Synechococcus* the new genus *Parasynechococcus*. Komarek et al. (2020) [34] divided *Synechococcus*-like species into several distinct, genetically and taxonomically separated lineages (genera) based on phylogenetic analyses. Based on several genome-level analyses, Salazar et al. (2020) proposed a new taxonomic framework that classified the genus *Synechococcus* as polyphyletic at the order rank and suggested that it may refer to 15 genera, which are placed into five distinct orders within the phylum Cyanobacteria: Synechococcales, Cyanobacteriales, Leptococcales, Thermosynechococcales, and Neosynechococcales [35]. The uncertainty of the phylogenetic systematics of Synechococcales leads to the instability of the nomenclature of species and strains of these cyanobacteria. For this reason, we will use “*Synechococcus*” throughout the text to refer the revealed “*Synechococcus*-like” picocyanobacteria.

To investigate the genetic diversity of *Synechococcus* in the Kara Sea, we used the *petB* gene as a phylogenetic marker and followed the “subclusters” classification and the genus name as it appears in the GenBank. The *petB* gene encodes the cytochrome b6 subunit of the cytochrome b6/f complex and shows a high taxonomic resolution for PC [30].

This study was conducted to determine the spatial distribution and abundance of picocyanobacteria in late autumn and trace the geographical origin of PC in the Kara Sea by identifying the lineage composition, using the *petB* gene as a marker.

## 2. Results

### 2.1. Environmental Parameters and Chlorophyll “a” Concentration

Map of the sampling sites is presented at Figure 1, the physical, chemical, and biological characteristics of each sampling station are given in Table 1. Sea surface temperature and salinity ranged from −1.7 to 2.0 °C and from 25.6 to 33.2, respectively, with the warmest temperature and lowest salinity recorded at southern stn 7505. The trend of decreasing salinity and increasing temperature towards the southern stations was observed throughout the 0–40 m layer (Appendix A). The euphotic depth varied from 20 to 64 m. All nutrients were depleted in the upper 20–30 m of the Transect except for stns 7501–7505, where Si(OH)_4_ concentrations exceeded 2 μM. Analyses of the DIN–PO_4_ molar ratio at different depths (0–100 m) showed that values were below than the Redfield value of 16:1 [36] and ranged from 0.2 to 15.6. The Redfield value was above 16:1 at depths less than 30 m only at northern stn 7494. Therefore, dissolved inorganic nitrogen was the macronutrient in the lowest supply for phytoplankton growth in the Kara Sea in autumn.

In the upper 50 m, Chl_tot_ varied from 0.06 to 0.24 mg m^−3^ and Chl_pico_ ranged from 0.02 to 0.05 mg m^−3^. The contribution of Chl_pico_ to Chl_tot_ varied significantly along the Transect, ranging from 16% to 53% (average 35%). The highest contribution of picophytoplankton (69%) occurred at a depth of 25 m, at st. 7499.

### 2.2. Cyanobacteria Abundance, Biomass, and the Contribution to Total Picophytoplankton

Cyanobacteria were not uniformly distributed over the upper 50 m. At northern stns 7494–7498 and 7500, *Synechococcus* was only detected in one or two horizons out of four or five. It was not revealed at stn 7499. At the southern stns 7501–7505, cyanobacteria were found in all sampled horizons. Within the upper 50 m, cyanobacterial abundance ranged from 2 to 88 cells mL^−1^. Biomass varied significantly from 0.043 to 2.07 mg C m^−2^, with the highest values found at stn 7505 (Figure 2, Appendix A). The lowest *Synechococcus* biomass was detected in the central and northern areas of the sea near the ice edge. *Synechococcus* abundance increased slightly, only at the northernmost stn 7494, and increased significantly towards the south. The contribution of *Synechococcus* to the total picophytoplankton biomass was low, reaching only 16% at the southernmost stn 7505.

### 2.3. Synechococcus Phylogenetic Diversity

DNA sequences from 17 samples were processed, and a total of 464,490 reads were obtained; 75.9% of these reads (352,519) were clustered and assigned to 100 *petB* operational taxonomic units (OTUs). Their sequences and occurrence in samples from explored stations are presented in Appendix A.

The alignment of these 100 sequences with 385 columns has 170 distinct patterns, 127 parsimony-informative positions, 52 singletons, and 205 constant sites. The base frequencies of sequences in the set are as follows: A: 0.170, C: 0.285, G: 0.253, and T: 0.292.

In the haplotype median-joining network (MJN), OTUs are grouped into nine clades (Figure 3) that correspond to subclusters on the ME phylogenetic tree (Figure 4).

Joint analysis of the representatives of these clades with sequences from GenBank in the ML phylogenetic tree in Figure 5 allows OTUs to be attributed to *Synechococcus* SC 5.1 and 5.2 (according to Herdman et al., 2001 [40] and Coello-Camba et al., 2023 [41]). The overall average *p*-distance for 100 OTUs is 0.088. The mean evolutionary divergence of OTUs within and between clades is presented in Table 2. Thus, the least variable is clade H and the most diverse is clade F of SC 5.2.

The alignment of *petB* sequences from 20 indicative OTUs and 32 GenBank accessions consists of 385 positions with 157 distinct patterns. There are 131 parsimony-informative positions, 21 singletons, and 233 constant sites in the alignment. The low value of *D* equal to −0.062073 in Tajima’s neutrality test indicates the absence of selection pressure in the diversity of OTUs.

On the constructed ML phylogenetic tree (Figure 5), the OTUs are divided into two main groups corresponding to *Synechococcus* SC 5.1 (87%) and 5.2 (13%). The vast majority of OTUs from SC 5.1 belong to lineage 5.1-I (85%). On the ME tree, the same main groups are revealed, but the bootstrap support is lower.

The most abundant SC, 5.1-I, was represented by 74 OTUs, mostly related to two *Synechococcus* phylotypes previously discovered in the eastern part of the Fram Strait (KX346155 and KX346036). These two phylotypes amount to 95% of the total reads of SC 5.1-I. Other *Synechococcus* phylotypes of SC 5.1-I (for example, KU377786, KX346059, and KF443068) were less represented in the community, with average proportions of reads below 1%.

SC 5.1-IV was represented by 13 OTUs, similar to phylotypes previously discovered in the Atlantic Ocean (KU705451) and Fram Strait (KX346152), and was found in almost all samples, contributing up to 7% of the total *petB* reads.

*Synechococcus* SC 5.2 (13 OTUs) was assigned to three phylotypes, and showed large differences in occurrence between samples, from being almost absent to increasing up to 5% of the total *petB* reads. Two phylotypes were closely related to *Synechococcus* sp. CP060395 from the winter plankton of Chesapeake Bay (clade G in Figure 4), CP075523 from the summer plankton of Narragansett Bay, and OY986430 from the Baltic Sea (clade H in Figure 4). The third *Synechococcus* phylotype, presented by OTUs 0008, 0054, 0031, and 0122 from SC 5.2 and discovered in almost all samples, was only moderately related to the sequences reported from alpine Lake Mondsee, Austria (LC716304).

The vertical distribution of all *Synechococcus* taxa was uniform, and no significant changes in depth were observed (Figure 6). The *Synechococcus* community structure at the surface appears to be generally representative of the community at greater depths. SC 5.1 dominates at all horizons across the Transect, followed by SC 5.2.

The association between cyanobacterial abundance and environmental parameters showed that *Synechococcus* increased with increasing water temperature (r_S_ = 0.65, *p* < 0.001) and silica (r_S_ = 0.63, *p* < 0.001); the weak positive correlation was found with Chl_tot_ and Chl_pico_ (0.35 and 0.28 *p* < 0.001) (Appendix A). Cyanobacterial abundance negatively correlated with salinity (r_S_ = −0.58, *p* < 0.001), dissolved inorganic nitrogen (−0.34 *p* < 0.05), and latitude (−0.62 *p* < 0.001). No clear relationship was found for phosphorus. Correlation analysis showed that the proportion of the most abundant *Synechococcus*, SC 5.1.I, negatively correlated with latitude and salinity. For SC 5.2, the proportion was negatively correlated with Si and PO_4_ concentrations. No significant trends were observed for *Synechococcus* SC 5.1.IV: the proportion of reads did not vary with latitude, temperature, salinity, nutrients, total chlorophyll “a”, or depth.

## 3. Discussion

In the present study, the spatial distribution of PC abundance and, for the first time, the phylogenetic diversity of *Synechococcus*, were investigated in the Kara Sea. The study was carried out in late September–early October 2022, covering the period before the formation of seasonal ice. Daylight duration did not exceed 10 h, and nitrate and phosphate concentrations were below the limiting values for phytoplankton growth, which at low temperatures are ~1 µM/L and 0.5 µM/L, respectively [42,43]. Silicon limitation (<2 µM/L) was found at stns 7494–7500 [44]. The concentrations of total Chl *a* were very low, characterizing the end of phytoplankton seasonal succession.

Cyanobacterial abundance corresponded to that obtained during three previous expeditions, with abundances ranging from 2 to 240 cells mL^−1^ of *Synechococcus* in the Kara Sea (Table 3). Our data was generally lower than that of *Synechococcus* (600 ± 250 cells mL^−1^) in autumn in the cold surface polar waters northwest of Svalbard (<2 °C, upper 50 m) [5]. The highest abundance was found in the southern part of the Transect at lower salinity and higher surface temperature and silicon concentration. These environmental conditions are determined by the Ob–Yenisei plume, which forms in the summer and remains in the central part of the Kara Sea until the end of the ice-free season. Autumn-specific wind conditions, namely, strong and prolonged southwesterlies, could move a part of this water eastward to the Laptev Sea [45]. As in other Arctic seas (Laptev and Beaufort), the maximum cyanobacterial abundance is restricted to areas influenced by river discharges, where warmer temperatures support the rapid growth rate of *Synechococcus* [46], with minimum abundance recorded further north in colder, saltier waters. (Table 3). Our results agree with previous studies identifying temperature and salinity as major environmental factors controlling the distribution and composition of *Synechococcus* [26,31,47].

The diversity of cyanobacteria belonging to the genus *Synechococcus* in the studied samples is limited mainly to SC5.1-I and 5.2; within these subclusters, the OTUs are diverse. Available data [34] indicate that the *Synechococcus* s.l is in fact a heterogeneous ensemble of phylogenetically diverse lineages that should be classified as different species or even genera. As stated in Figure 4 and Table 2, differences between clades A–E (SC5.1-I), I (SC5.1-IV), and F with H (SC5.2) significantly exceed OTUs’ dissimilarity within clades, amounting to more than 15% or 28–45 mutational steps, which would correspond to differences between species, although, of course, phylotypes identified via metagenomic analysis cannot be formally assigned to new taxa.

*Synechococcus* assemblages dominated by SC 5.1 lineage I, SC 5.1-IV, which frequently co-occurred with lineage I, were also observed at low relative abundance [54]. Both lineages inhabit high latitudes in cold and coastal waters [5,6,7]. Two OTUs (0012 and 0034) are close to accessions from the Atlantic and Arctic Oceans, referred to as belonging to SC 5.1 lineage IV. All revealed phylotypes had been previously found in Arctic regions—the North Sea, north Atlantic waters near Svalbard, and the Fram Strait. Considering the major routes of modified Atlantic waters (Figure 1, [55]) we can say that the eastern transport of north Atlantic waters is the main source of cyanobacterial inputs to the Kara Sea. Cottrell and Kirchman, 2009 [8], previously considered a similar hypothesis that photoheterotrophs, including coccoid cyanobacteria (*Synechococcus* and *Prochlorococcus*) found in Arctic coastal areas, are transported via advection from the Pacific Ocean to the Chukchi Sea.

Surprisingly, SC 5.2 was poorly represented. Members of this group are composed of euryhaline strains, found in river-influenced coastal waters, such as the Chesapeake Bay [26,56,57], the Pearl River estuary [58,59], the estuaries of Siberian rivers Khatanga, Kolyma, and Indigirka [60], and in low-salinity areas such as the Baltic Sea [61]. *Synechococcus* SC 5.2 has a wider thermal and salinity niche than other lineages [58], so we expected a higher phylogenetic diversity and relative contribution to *petB* reads, especially at the southern stations of the Transect influenced by river discharge. We assume that this is related to the late autumn sampling time. Most of the annual discharge in the Kara Sea, ~80%, occurs from May to the first half of September, after which it decreases significantly [38]. River water entering the sea is modified via mixing with ambient saline sea water, resulting in salinity exceeding 25 and sea surface temperatures barely reaching 2 °C, which is too low for *Synechococcus* SC 5.2 [62,63]. This leads to the death of most estuarine algae, a lack of net growth, and yet the survival of cyanobacteria. Overall, the occurrence of *Synechococcus* SC 5.2 in the area of the continental slope at 82° N provides evidence that its distribution may not be restricted to coastal areas. Previously, only *Synechococcus* SC 5.1 lineages I and IV were found up to 82.5° N, where they were also associated with the Atlantic inflow into the Arctic Ocean [5].

SC 5.2 is a genetically very diverse group [57], but clades within this group have still not been well defined [62]. Phylogenetic analysis showed that *Synechococcus* found in the Kara Sea and 95% related to the alpine phylotype LC716304 most likely refers to SC 5.2. Suzuki et al., 2022 [64] maintain that this alpine phylotype is not assigned and place it between Subalpine cluster 2 and SC 5.2. The environmental parameters under which this phylotype was discovered in the Kara Sea also suggest that it belongs to SC 5.2 and was brought into the sea through the river waters of Ob–Yenisei origin. The lack of a close relationship (<97%) with previously reported phylotypes could be due to the limited number of studies carried out in Arctic estuaries, so closely related sequences are yet to be deposited in GenBank. Alternatively, it could represent an unexplored species indigenous to the Kara Sea. Overall, we expect a higher number of undescribed *Synechococcus* in the Kara Sea, as these are the first such studies conducted in the region. With the exception of one phylotype described above, other revealed *Synechococcus* phylotypes were previously identified in the waters of Atlantic or river origin. It cannot be denied that such a low diversity of undescribed species is due to late autumn sampling, while a higher diversity would be expected in summer. We suppose that in summer, *Synechococcus* SC 5.2 may play a significant role in picoplankton. We have previously observed such a situation in the subarctic White Sea, where in winter, picophytoplankton containing only *Synechococcus* SC 5.1 entered the sea through Barents Sea waters, and in summer, the community was significantly richer and additionally represented by SC 5.2 and 5.3 [65].

The *Synechococcus* composition did not display vertical differentiation, as it was observed in Atlantic waters [7]. The *Synechococcus* assemblage showed similar compositions in the surface and deeper layers. Our samples for molecular analyses were restricted to the depth of the Surface Water Mass, and wind mixing increasing in autumn and the advection of Atlantic water evidently contribute to the uniform distribution of PC communities. Our results confirm a fairly uniform lineage composition at different depths previously obtained in the North Pacific, where the *Synechococcus* community structure at the surface is representative of the communities at depth [54].

In about 60% of the samples, the molecular data confirmed the luminescence microscopy results. At northern stations 7494, 7495, and 7596, where cyanobacteria were not detected at some sampled horizons, molecular analysis revealed their presence. This could be due to several factors. First, the volume of the subsample for microscopy was only 20 mL, and given that there was a low number of cells in the sample, they may not have been included in the analysis. Second, the water for environmental DNA analysis was passed through 0.2 µm filters, which can collect not only living organisms, but also DNA from dead cells. Dissolved extracellular DNA is ubiquitous in marine environments [66] and can range from a several hundred to several thousand base pairs in length [67]. Therefore, the PC sequences detected in the samples may represent dead cells.

## 4. Materials and Methods

### 4.1. Water Sample Collection

Seawater samples were collected during the 89th cruise of the R/V “Akademik Mstislav Keldysh” on 30 September–3 October 2022. A total of eleven stations (stns) were sampled along a Transect running from the north to the southeast of the Kara Sea (Figure 1). Seawater was sampled directly from a 5 L Free Flow Water Sampler (Hydro-Bios Inc. Kiel, Germany) mounted on a SBE 32 Carousel Water Sampler (Sea-Bird Scientific Inc., Bellevue, WA, USA) with a SBE 911plus CTD (conductivity, temperature, and pressure), and sensors for chlorophyll fluorescence (ECO Chlorophyll Fluorometer, Wet Labs Inc., Bellevue, WA, USA). The Free Flow Water Sampler is an innovative design that avoids the limitations of the PWS/Niskin bottles due to their reduced inlet and outlet diameters. Salinity is reported using the EOS-80 Practical Salinity scale. Water samples were collected from several horizons: the surface (0.5–1 m), the halocline, and under and below the halocline (49 samples in total, see Appendix A).

The samples (total 17) for genetic analysis were collected at eight out of the eleven stations from the surface and the halocline (stns 7494–7496, 7498, 7500, 7501, 7503, and 7505, Figure 1).

The northernmost station (stn) of the Transect, 7494, was located near the edge of multi-year ice; its maximum northern seasonal position during the study period (82°15.7′ N) allowed for observations in the area of the continental slope at depths of up to 1650 m in a virtually unexplored region of the Kara Sea. Stations 7498 and 7499 were located near the Western edge of the Severnaya Zemlya ice massive (https://worldview.earthdata.nasa.gov (accessed on 5 February 2025)).

Water masses were defined based on vertical profiles of temperature, salinity, and density. According to the criteria presented by Cokelet et al., 2008 [68], the depth of Surface Water Masses with a salinity of <34.9 and a density of <27.7 kg m^−3^ varied from 35 m to 90 m at stns 7494–7502, while Arctic Water Masses with a salinity < 34.9 and density > 27.7 kg m^−3^ were located deeper. Surface water spread to the bottom at stns 7503–7505. In autumn 2022, the Surface Water Masses were mainly composed of freshened waters from river runoff in the central area of the Kara Sea and modified Atlantic Waters in the northern area of the sea [69,70,71].

### 4.2. Nutrients and Chlorophyll a Concentration

Nitrate (NO_3_^−^), nitrite (NO_2_^−^), ammonium (NH_4_^+^), phosphate (PO_4_^3−^), and silicate (Si(OH_4_)) concentrations were measured on board the ship in the few hours after sampling. The content of dissolved inorganic phosphorus was measured using the modified Morphy and Riley method. Dissolved inorganic silicon was estimated via colorimetric analysis using a blue silicon–molybdenum complex (Korolev’s method); nitrite nitrogen was measured via calorimetry using a single-color reagent, as was nitrate nitrogen after its reduction to nitrite in cadmium columns. Ammonia nitrogen was determined via the Sadgi–Solorzano method and the phenolate-hypochlorite reaction. The methods were described in detail in manuals [72,73]. For calorimetry, we used a Lange Hach 2800 or 3900 spectrophotometer in a single-wavelength mode. All nitrogen forms were then summed and labeled as dissolved inorganic nitrogen (DIN).

Water samples (500 mL) for total Chl-*a* concentrations (Chl_tot_) were filtered through Whatmann GF/F filters (47 mm). For the pico-sized fractionated Chl-*a* concentrations (Chl_pico_), 1 L water samples were passed through a 3 μm-pore-size polycarbonate filter using a <50 mmHg vacuum. The filtrates (<3 μm) were then filtered through 0.7 µm Whatmann GF/F filters (47 mm). Chl-*a* was measured using a Trilogy field fluorometer from the Turner Designs model, calibrated spectrophotometrically using a chemically pure chlorophyll solution (Sigma, St. Louis, MO, USA) as a standard. The Chl-*a* and phaeophytin concentrations were calculated according to Holm-Hansen and Riemann, 1978 [74].

### 4.3. Total Picophytoplankton and PC Abundance and Biomass

The abundance of picoeukaryotes and cyanobacteria was estimated using epifluorescence microscopy. Sub-samples (10 mL) were placed in a filtration funnel and incubated for 5–7 min after the addition of a saturated solution of primulin. Each sample was preserved with glutaraldehyde at a final concentration of 1%. Nuclear filters (0.2 μm pore diameter, Dubna, Russia) prestained with Sudan black were used for filtration. Cells on the filter were counted under a Leica DM1000 epifluorescence microscope at a ×100 ×10 ×1.3 magnification. Depending on the cell concentration, 30–50 fields were examined, and cell size was measured. The “type” of fluorescence was also determined: spherical cells with a diameter of ≤1.5 μm with orange fluorescence from phycoerythrin (575 ± 20 nm) were considered to be PC. Orange fluorescence under blue excitation was also peculiar for cryptophytes, but the latter can be easily identified by their asymmetric cell shape and were absent in our samples. Cell volume was converted to carbon using different conversion factors. For prokaryotic cells with sizes ranging from 0.8 to 1.2 µm (average 1 µm), a conversion factor of 470 ƒg C/cell was used [75]. The carbon biomass of picoeukaryotes was estimated using the conversion factor logC = 0.941logV − 0.60 [76], where V is the cell volume calculated as the volume of the relevant geometric bodies [77]. Integral biomass values based on the trapezoid rule were calculated at depths from 0 to 50 m at each station in this study.

### 4.4. Determination of Euphotic Depth

The intensity of surface and underwater irradiance measured using LI-190SA and LI-192SA (LI-COR) sensors, respectively, was used to estimate the depth of the euphotic zone (Z_eu_, 1% of surface irradiance). In the absence of underwater hydro-optical measurements, the diffuse attenuation coefficient for downwelling solar radiation in the visible spectrum was calculated according Demidov et al., 2017 [78].

### 4.5. DNA Extraction, PCR Amplification, Cloning, and Sequencing

Seventeen samples were analyzed to investigate the PC diversity. For DNA isolation, 3–5 l of water sample was filtered through a 3 µm-pore-size polycarbonate filter using a <50 mm Hg vacuum. The filtrate (<3 µm) was then filtered through 0.2 µm Sterivex units (Millipore Canada Ltd., Mississauga, ON, Canada). The buffer (1.8 ml of 50 mM Tris–HCl, 0.75 M sucrose, and 40 mM EDTA; pH 8.3) was added to the Sterivex units and the samples were stored at −80 °C until nucleic acid extraction. The total DNA was extracted from the filters as described [79].

The *petB* gene is a single copy, which reduces the amplification bias. The sequences of this gene are easily aligned, as they are conservative in length, amounting to 657 bp. Primers *petB*-F and *petB*-R, proposed by Mazard et al., 2012 [30], are complementary to the most conserved regions of the gene, and produce a 597 bp amplicon. This length is too long for NGS sequencing. We used the primers to generate a 425 bp-long gene fragment that is optimal for 2 × 250 bp pair-end sequencing on the Illumina platform. The *petB*-F (5′-TACGACTGGTTCCAGGAACG-3′) from Mazard et al., 2012 [30], was used as a forward primer. The reverse primer *petB*-587r (5′-CARTARCCAACYTGRTCCCA-3′) [41] was degenerated and produced a non-specific PCR product in controls without a forward primer. The number of non-specific bands increased at annealing temperatures of 56° and below, while at temperatures above 58°, the intensity of the band of the expected size decreased significantly, along with a decrease in non-specificity. As a result, the primer’s annealing temperature was set at 58° to prepare libraries for NGS sequencing. PCR was performed using the Encyclo Plus PCR kit (Evrogen, Russia) reagents. The PCR conditions were as follows: initial denaturation at 95° for 3 min., followed by 35 cycles of denaturation at 95° for 20 s, annealing at 58° for 20 s, and elongation at 72° for 40 s, with a final elongation at 72° for 5 min. Amplicons of the expected size were excised from 1% agarose gels and purified using the Cleanup-mini kit (Evrogen, Moscow, Russia).

The library preparation and sequencing of the DNA fragments were carried out using the TruSeq Nano DNA Kit according to the manufacturer’s protocol, using the Illumina NovaSeq 6000 (San Diego, CA, USA). The read length was 250 bp, and reading was performed from both sides of the fragments. Sequence reads were processed with DADA2 on primer-free reads to correct sequencing errors [80]. The reads were quality-filtered, dereplicated, and merged, and chimeras were removed according to the script (https://github.com/deniseong/marine-Synechococcus-metaB (accessed on 5 February 2025)). The OTUs with at least 97% sequence similarity were first checked taxonomically against a reference *petB* database [41], and all unclassified OTUs were then manually checked against the NCBI nucleotide database using BLAST (version BLAST+ 2.16.0.)

The sequences obtained from the study were deposited in GenBank (http://www.ncbi.nlm.nih.gov (accessed on 1 May 2025)) under accession numbers PV433308-PV433317, PV575870-PV575970, and BioProject PRJNA1231002.

### 4.6. Phylogenetic Analysis

The nucleotide sequences were aligned manually in BioEdit 7.2.5 (https://bioedit.software.informer.com/ (accessed on 17 April 2025)). The Minimum Evolution (ME) method [81] was used for tree construction in MEGA ver.12.0.9 [82] using 1000 bootstrap replicas. The evolutionary distances in the ME tree were computed using the Maximum Composite Likelihood method [83] and were measured in units of the number of base substitutions per site. The rate variation among sites was modeled with a gamma distribution (shape parameter = 1). The differences in the composition bias among sequences were considered in evolutionary comparisons.

A Maximum Likelihood (ML) tree was inferred using IQ-TREE ver.2.3.6. (http://www.iqtree.org) [84], with the GTR+F+G4 substitution model, determined as best fitting by the Akaike Information Criterion using the ModelFinder [85] implemented in IQ-TREE. The rate variation among sites was modeled with a gamma distribution (shape parameter = 1). Ultrafast bootstrap analyses [86] were performed using 1000 replicas.

*p*-distances for sequences and Tajima’s neutrality test [87] were calculated in MEGA. The haplotype median-joining network (MJN) [88] was constructed in the POPART 1,7 software package [89] (https://popart.maths.otago.ac.nz/download/ (accessed on 17 April 2025)).

### 4.7. Statistical Analysis

Spearman’s rank correlation was applied to study the relationships between variables. The influence of environmental factors on PC abundance and SC read proportions was estimated using PAST 3.10 [90].

## 5. Conclusions

Although the abundance of picocyanobacteria was low in the autumn, the picophytoplankton of the Kara Sea contained two main *Synechococcus* subclusters: SC5.1 lineage I, SC5.1 lineage IV, and SC5.2. Both SC5.1 lineages are marine in origin and inhabit high latitudes in cold and coastal waters, whereas SC5.2 is derived from freshwater. All found subclusters can coexist and survive the cold autumn season in the Arctic Sea. Our study showed that temperature and salinity are the main environmental factors controlling the distribution and composition of *Synechococcus* in the Kara Sea. When temperatures are low and river discharge decreases, *Synechococcus* SC5.1 dominates in the PC communities. The *petB* gene revealed the genetic diversity within Synechococcus populations in the Kara Sea in autumn, revealing novel genetic sequences that have not been recognized before. Further study of Kara Sea picophytoplankton in other seasons is necessary to gain a more complete picture of the PC composition. Modeling predicts that the maximum warming in the Eurasian sector of the Arctic Ocean over the past four decades has occurred in the north of the Barents and Kara Seas near Svalbard and Novaya Zemlya [91]. Therefore, cyanobacteria are expected to become increasingly important members of picophytoplankton as the Atlantic water inflow and river discharge increase.

## Figures and Tables

**Figure 1 plants-14-02614-f001:**
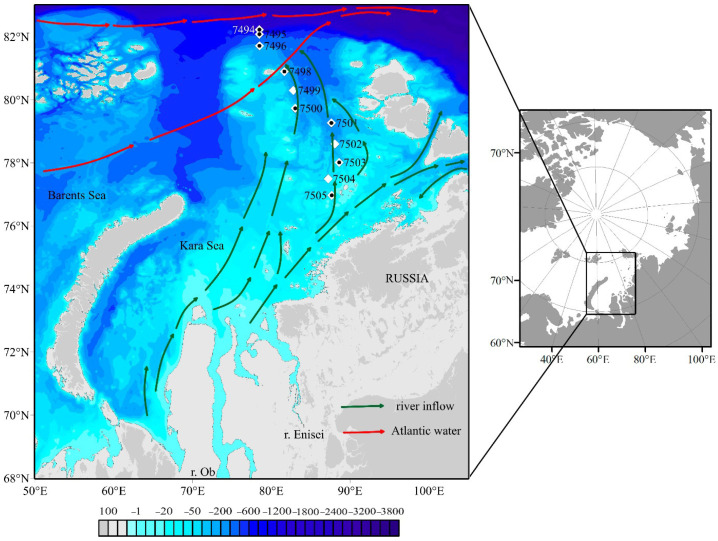
Map of the sampling sites (white squares). Black dots indicate stations where samples for molecular analysis were collected. Bathymetry is given according to IBCAO https://www.gebco.net/data-products/gridded-bathymetry-data/arctic-ocean (accessed on 17 April 2025). The scheme of Atlantic Water Masses and river inflow is given according www.amap.no and [37,38,39].

**Figure 2 plants-14-02614-f002:**
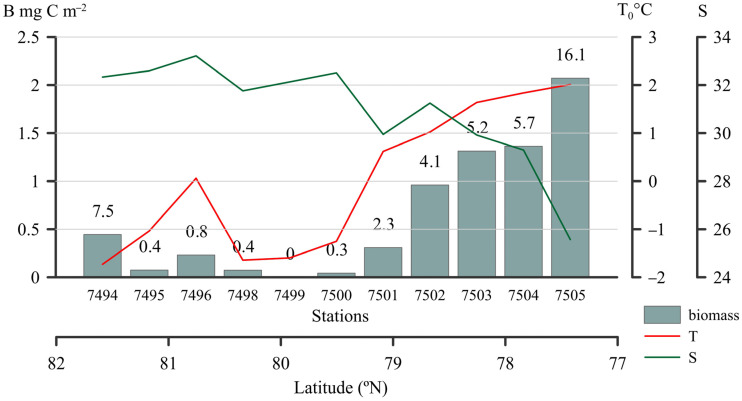
Integrated picophytoplankton biomass (B, mg m^−2^), temperature (T_0_ °C), and salinity (S_0_) in the surface layer. Numbers—contribution of cyanobacteria to the total picophytoplankton biomass (in %).

**Figure 3 plants-14-02614-f003:**
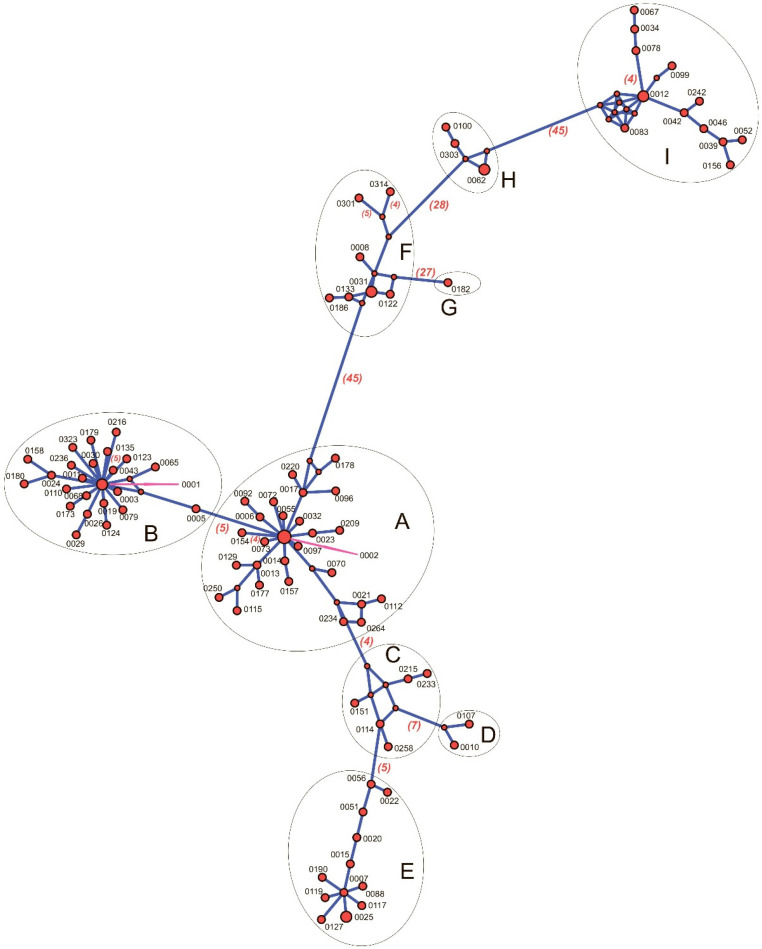
Median-joining network of 100 OTUs in clades. The number of mutation steps that are more than three is shown. The defined clades A–H are indicated.

**Figure 4 plants-14-02614-f004:**
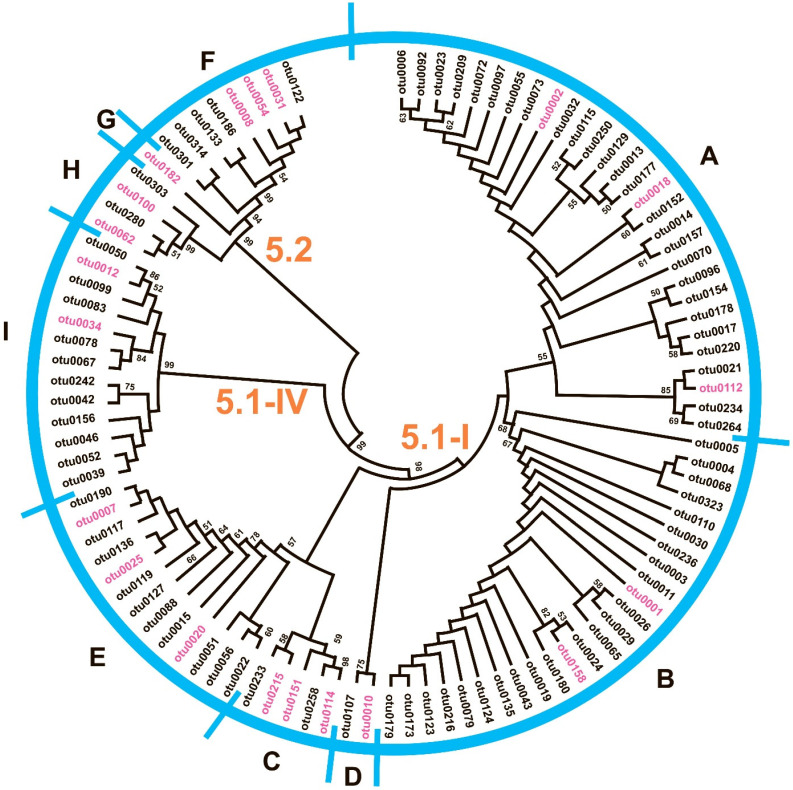
ME tree for 100 OTUs. OTUs included further in the ML phylogenetic tree in Figure 5 are highlighted in purple. The assignment of OTUs to clades corresponded to subclusters in the median-joining network in Figure 3 is indicated. The clades A–H are indicated.

**Figure 5 plants-14-02614-f005:**
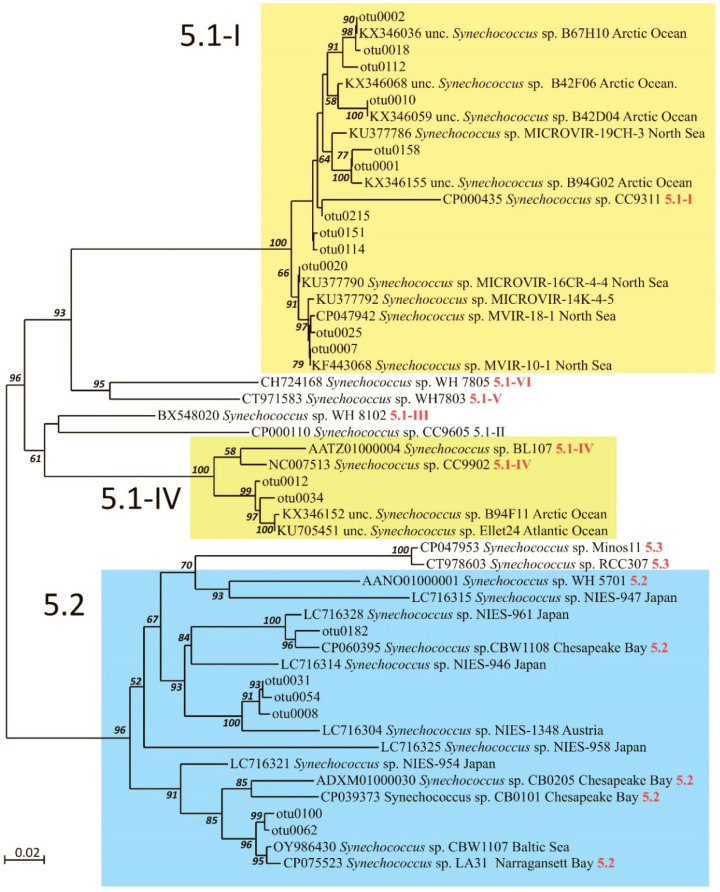
ML phylogenetic tree based on *Synechococcus petB* sequences. Ultrafast bootstrap support values > 50% are indicated. The scale bar represents the number of substitutions per site.

**Figure 6 plants-14-02614-f006:**
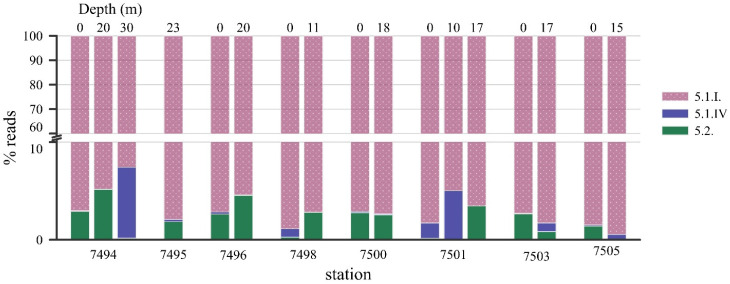
*Synechococcus* assemblage compositions (% reads) assigned to subclusters, organized by station and depth.

**Table 1 plants-14-02614-t001:** H—station depth (m); Z_eu_—depth of the euphotic layer (m); T_0_ °C and S_0_—temperature and salinity in the surface layer; DIN, PO_4_^3–^, and Si(OH)_4_—concentrations of the sum of nitrite and nitrate nitrogen (DIN), phosphates, and dissolved silicon in the surface layer, respectively (μM L^−1^); and Chl_tot_ and Chl_pico_—the average concentrations of Chl_tot_ and Chl_pico_ in the 0–50 m layer (mg m^−3^).

Station	H	Z_eu_	Surface					Chl_tot_	Chl_pico_
			T_0_ °C	S_0_	PO_4_^3–^	Si(OH)_4_	DIN		
7494	1652	40	−1.7	32.3	0.09	0.46	0.49	0.15	0.02
7495	490	29	−1.0	32.6	0.08	0.20	0.22	0.24	0.05
7496	176	34	0.1	33.2	0.08	0.15	0.13	0.13	0.05
7498	196	43	−1.6	31.8	0.11	0.31	0.02	0.11	0.04
7499	233	47	−1.6	32.1	0.11	0.46	0.32	0.10	0.04
7500	83	64	−1.3	32.5	0.10	0.25	0.09	0.06	0.03
7501	292	20	0.6	29.9	0.13	3.54	0.05	0.13	0.05
7502	230	23	1.0	31.2	0.12	1.49	0.07	0.14	0.05
7503	104	21	1.6	29.9	0.10	3.13	0.05	0.13	0.05
7504	120	22	1.8	29.3	0.14	5.37	0.1	0.13	0.05
7505	72	24	2.0	25.6	0.16	11.99	0.02	0.09	0.03

**Table 2 plants-14-02614-t002:** Mean *p*-distances over OTUs pairs between and within clades.

	A	C	B	D	E	I	G	F	Within Clades	Number of OTUs
A									0.012	29
C	0.029								0.009	5
B	0.031	0.030							0.011	25
D	0.034	0.033	0.036						0.010	2
E	0.040	0.025	0.047	0.041					0.009	13
I	0.166	0.161	0.164	0.160	0.168				0.013	13
G	0.169	0.177	0.179	0.174	0.170	0.173			-	1
F	0.170	0.177	0.175	0.178	0.173	0.175	0.098		0.017	8
H	0.174	0.174	0.176	0.170	0.162	0.156	0.126	0.111	0.006	4

**Table 3 plants-14-02614-t003:** Cyanobacteria abundance (N, cells mL^−1^) and the contribution (%) to the total picophytoplankton biomass in Arctic estuaries and seas.

Data	Location	Cells mL^−1^	%	Reference
Kara Sea				
September 2017	Shelf, Ob estuary	190–240	0.2–11	[48]
September 2017	Northwest	20	–	[49]
July 2019	Northwest	20–70	–	[50]
June 2021	Western area	8–16	0.2–2	[51]
October 2022	Shelf, continental slope	2–88	0.3–16	This study
Laptev Sea				
September 1991	Lena River delta	1000–30,000	–	[52]
September 2017	Continental slope, shelf, Khatanga estuary	20–1250	1–26	[53]
Beaufort Sea				
September 2002	Shelf, Mackenzie estuary	215–2400		[22]

## Data Availability

The data generated or analyzed during this study are included in this published article and its Appendix A. Sequences obtained from this study were deposited in GenBank (http://www.ncbi.nlm.nih.gov (accessed on 5 May 2025)) under accession numbers PV433308-PV433317, PV575870-PV575970, and BioProject PRJNA1231002.

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
