# Peer review of "Distribution and Phylogenetic Diversity of Synechococcus-like Cyanobacteria in the Late Autumn Picophytoplankton of the Kara Sea: The Role of Atlantic and Riverine Water Masses"

_plants, 2025, doi:10.3390/plants14172614_

Round 1
Reviewer 1 Report
Comments and Suggestions for Authors
I understand the samples were analyzed during a research cruise, and unique temporal representation (late autumn) of the area was obtained: justify the relevance (or characteristic) of the area in this season. Why is autumn important?
Results:
You indicated only some values in some depths. It is better to show a summary of all values (you have 11 stations, 4 samples at different depths, so must have at least 44 values). I could not see supplementary figures or tables. Anyway, you cited that as important information, so you should include as a part of the results.
Figure 2. Indicate with precision the meaning of the elements in the graph: bar-biomass, line red-temperature, line green-salinity, x-axis stations, etc.
Figure 6. Until I saw this result, I understood the number of your samples that did not match what you declared in methods: 8 stations, 2 depths (total 16). You have different depths in the stations. Explain well in methods.
The conclusion is not such. You are inferring beyond the season you investigated. Please conclude appropriately based on the facts you have.
Author Response
Reply to Reviewer #1
We are extremely grateful to the Reviewer for the careful analysis of the manuscript and valuable comments which allowed us to improve the text of the article. We have prepared a revised version of the paper taking into account the Кeviewer’s comments.
Reviewer point 1:
I understand the samples were analyzed during a research cruise, and unique temporal representation (late autumn) of the area was obtained: justify the relevance (or characteristic) of the area in this season. Why is autumn important?
Response 1: As mentioned in the introduction, the Arctic seas, including the Kara Sea, are difficult to access. Large masses of melting ice block the Kara Gate Strait until early August. September is the most convenient month for conducting research, as the sea is almost free of ice. Moreover, the abundance and biomass of micro- and nanophytoplankton decrease in autumn, resulting in an increased contribution of picophytoplankton — particularly cyanobacteria — to the total biomass and chlorophyll in the Kara Sea (Demidov et al., 2021) [3]. In these conditions, the method of fractionated filtration used in our research produces fewer errors.
Reviewer point 2:
Results:
You indicated only some values in some depths. It is better to show a summary of all values (you have 11 stations, 4 samples at different depths, so must have at least 44 values). I could not see supplementary figures or tables. Anyway, you cited that as important information, so you should include as a part of the results.
Response 2:Picophytoplankton biomass is presented as integral biomass values based on the trapezoid rule calculated depths from 0 to 50 m at each station (see Mat&Mat, paragraph 2.3) at Figure 2. According the Reviewer note we added primary data of abundance and biomass in Table S1 of Supplemental Materials.
Reviewer point 3:
Figure 2. Indicate with precision the meaning of the elements in the graph: bar-biomass, line red-temperature, line green-salinity, x-axis stations, etc.
Response 3:It was corrected in the revised version of the manuscript.
Reviewer point 4:
Figure 6. Until I saw this result, I understood the number of your samples that did not match what you declared in methods: 8 stations, 2 depths (total 16). You have different depths in the stations. Explain well in methods.
Response 4: Samples for PC abundance and biomass were collected at 11 stations from different horizons, total 49 samples were analyzed (see Mat&Mat, paragraph 2.1). Samples for genetic analysis (total 17) were collected at eight out of the eleven stations from the surface and the halocline (see Mat&Mat, paragraphs 2.1 and 2.5).
Reviewer point 5:
The conclusion is not such. You are inferring beyond the season you investigated. Please conclude appropriately based on the facts you have.
Response 5: We rewrite Conclusion according to Reviewer point:“In autumn, when temperatures are low and river discharge decreases, the Arctic phylotypes Synechococcus SC 5.1 dominate in the PC communities of the Kara Sea. At the same time, the intensity of the influx, as well as growth rate and distribution of cyanobacteria, depend on the season. We suppose that in summer this picture may change significantly and Synechococcus SC 5.2 may play significant role in picoplankton. We have previously observed a such situation in the subarctic White Sea where in winter, picophytoplankton contained only Synechococcus SC 5.1, entered the sea with the Barents Sea waters, and in summer, the community was significantly richer and additionally represented by SC 5.2 and 5.3 [73]. The petB gene discovers genetic diversity within Synechococcus populations in the Kara Sea in autumn, revealing novel genetic sequences that have not been recognised before. It is necessary to continue studying the Kara Sea picophytoplankton in other seasons to get a most complete picture of the PC composition. Modeling predicts that the maximum warming in the Eurasian sector of the Arctic Ocean over the past 4 decades has occurred in the north of the Barents and Kara Seas near Svalbard and Novaya Zemlya [88]. Therefore, cyanobacteria are expected to become an increasingly important member of picophytoplankton as Atlantic water inflow and river discharge increase.”
Reviewer 2 Report
Comments and Suggestions for Authors
This is a well-designed, well-conducted, and clearly written manuscript focusing on picophytoplankton, more precisely picocyanobacteria, in the Kara Sea. This northern region in general is receiving increasing scientific attention due to the accelerating impacts of global warming, which are rapidly transforming marine environments. The authors investigated the spatial distribution of picocyanobacterial abundance and the phylogenetic diversity of Synechococcus in the Kara Sea. They found that picocyanobacterial abundance increases with rising temperatures and decreasing salinity. The Synechococcus community was characterized using amplicon sequencing targeting the petB gene, which revealed the presence of Synechococcus subcluster 5.1 (polar lineages I and IV) and the euryhaline subcluster 5.2. Notably, subcluster 5.2 was documented for the first time as far north as 82°N.
I do not have major comments. It would be only beneficial to add few sentences in the introduction about the importance of picophytoplankton in general.
Author Response
Reply to Reviewer #2
Thank you very much for your appreciation of our work. We hope you found it interesting.
Reviewer point:
I do not have major comments. It would be only beneficial to add few sentences in the introduction about the importance of picophytoplankton in general.
Author response:
The following has been added to the Introduction of the revised version of the manuscript:
“Unicellular picocyanobacteria [<2 µm] belong to the functional class of picophytoplankton, the smallest algae which contribute substantially to both total phytoplankton biomass and primary production in marine ecosystems, especially in the oligotrophic waters of the Arctic Ocean, where they account for up to 55%–90% of the total photosynthetic biomass and carbon production [1-3].”
Reviewer 3 Report
Comments and Suggestions for Authors
This article is devoted to the phylogeography of picocyanobacterial genus Synechococcus from the plankton of the Kara Sea. The article is well designed and the text is clear. For studying the genetic diversity of Synechococcus the petB gene was used. The authors showed that Synechococcus from the Kara Sea belongs to the subclusters 5.1-I, 5.1-IV and 5.2.
This article is an important contribution to the study of picoplankton from the Kara Sea. The results can be used for further ecological monitoring and phylogeographic studies. I think the manuscript is suitable for publication in Plants and it can be accepted after the minor revision.
I have only a few comments.
- Are you sure that your sequences belong to the "true" Synechococcus? Relatively recently Komarek et al. (2020) revised the Synechococcus-like cyanobacteria and showed that marine isolates of Synechococcus belong to Parasynechococcus. By the way, please correct the reference of this article (line 636) https://fottea.czechphycology.cz/artkey/fot-202002-0009_phylogeny_and_taxonomy_of_synechococcus-like_cyanobacteria.php
- Please, add a few words about the division of Synechococcus to clusters/subclusters in the "Introduction" for better understanding.
- Lines 209-222: it is better to place these two paragraphs in the "Materal & Methods" section.
- Lines 360-361: "The diversity of cyanobacteria belonging to the genus Synechococcus in the studied samples is limited to 5.1-I, IV and 5.2" Please, add "subclusters".
In general, the English language is fine, however, its minor improvement would be useful. For instance, lines 333-334: "...the first phylogenetic diversity of Synechococcus were investigated in the Kara Sea" What is the first phylogenetic diversity? Do you mean: "...were investigated for the first time"?
Author Response
Reply to Reviewer #3
Thank you so much for forwarding comments on our manuscript. We have followed the valuable comments and suggestions, made corrections and addressed them thoroughly in the revised manuscript and below.
Reviewer point 1:
Are you sure that your sequences belong to the "true" Synechococcus? Relatively recently Komarek et al. (2020) revised the Synechococcus-like cyanobacteria and showed that marine isolates of Synechococcus belong to Parasynechococcus. By the way, please correct the reference of this article (line 636) https://fottea.czechphycology.cz/artkey/fot-202002-0009_phylogeny_and_taxonomy_of_synechococcus-like_cyanobacteria.php
Response 1: Thank you for your comment! We have certainly read the works of Komarek et al. (2020) and Coutinho et al (2016) where they identify new species within the polyphyletic genus Synechococcus. We used a petB gene database and data from a GenBank, where the sequences were annotated as Synechococcus. In addition, when comparing the results obtained with published data from other regions, the “old” nomenclature was preferable. We plan to continue our research on cyanobacteria in the Arctic seas and will take into account and apply the new species identified within Synechococcus–like populations.
The following phrase has been added to the Introduction: "Coutinho et al. (2016) [34] based on phylogenomic reconstruction segregate from Synechococcus the new genus Parasynechococcus. Komarek et al. (2020) [35] divided Synechococcus–like species into several distinct, genetically and taxonomically separated lineages (genera) based on phylogenetic analyses.
In order to investigate the genetic diversity of Synechococcus in the Kara Sea, we used the petB gene as a phylogenetic marker, the "subclusters" classification,.and the genus name as it appears in the GenBank."
The reference Komarek et al (2020) has been changed.
Reviewer point 2:
Please, add a few words about the division of Synechococcus to clusters/subclusters in the "Introduction" for better understanding.
Response 2: We added information about the division of Synechococcus into clades and subclusters in the "Introduction". We also describe the ecological preferences of the identified clades and subclusters in detail in the 'Discussion' section.
“Various molecular genes have been used to identify cyanobacteria such as 16S rRNA, ITS region, rpoC1, narB, ntcA, and petB genes [24-29]. Based on several genome-level analyses, Salazar et al. (2020) proposed a new taxonomic framework that classified the genus Synechococcus as polyphyletic at the order rank and suggested that it may refer to 15 genera, which are placed into five distinct orders within the phylum Cyanobacteria: Synechococcales, Cyanobacteriales, Leptococcales, Thermosynechococ-cales and Neosynechococcales [25]. Dufresne and coauthors (2008) analyzed the complete genomes of 11 marine Synechococcus isolates and classified them into three subclusters, 5.1, 5.2 and 5.3 [26]. Other authors analyzing different markers have proposed various nomenclatures for marine Synechococcus, recognising up to 27 clades within subclusters 5.1–5.3 [5]. Ahlgren and Rocap, based on multiple gene loci, recognized more than 30 marine cultured Synechococcus clades in three subclusters [29].”
Reviewer point 3:
Lines 209-222: it is better to place these two paragraphs in the "Materal & Methods" section.
Response 3: This has been changed in the revised version of the manuscript.
Reviewer point 4:
Lines 360-361: "The diversity of cyanobacteria belonging to the genus Synechococcus in the studied samples is limited to 5.1-I, IV and 5.2" Please, add "subclusters".
Response 4: This has been corrected in the revised version of the manuscript.
Reviewer point 5:
In general, the English language is fine, however, its minor improvement would be useful. For instance, lines 333-334: "...the first phylogenetic diversity of Synechococcus were investigated in the Kara Sea" What is the first phylogenetic diversity? Do you mean: "...were investigated for the first time"?
Response 5: Yes, you are right., we meant that “phylogenetic diversity of Synechococcus in the Kara Sea was investigated for the first time”. The sentence was changed in the revised version.
We have corrected grammatical and typographical errors in the revised version of the manuscript.